# Acceptance and Commitment Training Focused on Psychological Flexibility for Family Members of Children with Intellectual Disabilities

**DOI:** 10.3390/ijerph192113943

**Published:** 2022-10-27

**Authors:** David Lobato, Francisco Montesinos, Eduardo Polín, Saray Cáliz

**Affiliations:** 1Department of Psychology, Universidad Europea de Madrid, 28670 Madrid, Spain; 2Instituto ACT, 28036 Madrid, Spain

**Keywords:** parenting, psychological flexibility, disability, contextual therapies, acceptance and commitment therapy, ACT, parental stress

## Abstract

The objective of the study was to analyse the effect of a psychological flexibility intervention programme based on Acceptance and Commitment Therapy (ACT) on 36 family members of children with intellectual disabilities. The 6-PAQ (parental psychological flexibility), PSS-14 (perceived stress), GHQ-12 (psychological health), and WBSI (suppression of unwanted thoughts) were used as measurement instruments before the programme (pre), after (post), and at follow-up (after two months). Possible change in family interactions due to the family intervention was also assessed through self-monitoring. A decrease in psychological inflexibility, a reduction in stress, an improvement in psychological well-being, and a reduction in the tendency to suppress thoughts and emotions were observed after the programme. Furthermore, the effects seem to extend to family interactions, with an increase in positive interactions and a decrease in negative ones. The study leads us to think about the importance of psychological flexibility in children with chronic conditions as a process that mediates the impact of stress and family well-being.

## 1. Introduction

Parents of children with disabilities are at higher risk of psychological stress than other parents [1]. This stress appears to be mediated by many variables, such as the role of the caregiver [2], the coping styles [3] and the locus of control [4], the perceived social support [5], the professional support, guidance and supervision [6], the socio-economic status of the family [7] and the severities and impairments of the intellectual disability [8] among others. In addition, this stress can be manifested in health problems [9] and alterations in psychological functioning, such as depression, anxiety, and poorer quality of life [10], as well as ineffective parenting skills, higher levels of hostility, and lower parental responsiveness [11]. These psychological problems are aggravated by behavioural problems in diagnosed children [12]. This stress impacts indirectly on the psychological functioning of the diagnosed child [13], dramatically worsening family interactions and generating an aversive and hostile environment as the more problematic behaviour of the child diagnosed with a disability becomes [14]. This results in an authoritarian, harsh and intrusive upbringing [15], contributing to the worsening of the psychological health of the family members, especially children with disabilities. These individuals would, as a result, reach lower levels of development and global functioning [16]. Therefore, a family-centred therapeutic approach for children with disabilities seems to be unclear [17] because the behavioural repertoires of parents affect the children and the repertoires of the children influence the parents [18]. This can improve the family’s quality of life, their functioning in daily life and adjustment to different life scenarios [19,20], as well as the impact of the disability [21]. The COVID-19 health crisis has increased stress levels in the general population [22], being multiplied in the case of parents of children with disabilities [23], who are already vulnerable. This may be decreasing their health levels and interfering with the development of their children’s psychological well-being. Acceptance and Commitment Therapy (ACT) is extending its research into parenting, family, and chronic child problems such as disability, applying psychological flexibility (PF) training to improve psychological well-being, perceived stress, and family quality of life [24,25]. ACT is a third-wave behavioural approach that uses six core processes (acceptance, contact with the present moment, cognitive defusion, self-as-context, values, and committed action) to promote PF: understood as the process of noticing experiences in the present moment without judging them and persisting with or changing behaviours to serve valuable ends [26]. PF has been consistently presented as a predictor of long-term psychological well-being, especially in chronic stressors or problems [27].

In a previous study by our team, psychological flexibility was found to be related to the impact of parental stress and health issues associated with raising a child with a disability [28]. Likewise, a preliminary study showed the value of a protocol based on PF to reduce such psychological consequences in a group of 5 family members [29]. Therefore, this study evaluated the usefulness of a psychological flexibility-oriented intervention protocol in a group format in a larger group of family members of children with disabilities. The aims were to reduce stress, improve quality of life, and modify maladaptive interaction repertoires among family members by incorporating some improvements in the protocol and collecting data more extensively than in the previous study.

## 2. Materials and Methods

### 2.1. Design

After obtaining permission from the European University ethics committee, 40 family members (parents) were recruited to participate in an open clinical trial from an association of intellectual disability in the Community of Madrid. The inclusion criteria were to be over 18 years of age, in the absence of current psychological and/or psychiatric treatment, with a high level of understanding of Spanish, and to have children diagnosed with intellectual disability. A total of 4 family members did not continue the study or dropped out (2 due to incompatibility of schedule with the programme, 1 due to health problems, and 1 for unknown reasons). Four interventions were carried out in different groups (group 1 = 10 participants; group 2 = 10 participants, group 3 = 10 participants, group 4 = 6 participants) so that when the programme ended with one group, the next one started.

### 2.2. Participants

Most of the relatives were female (83.34%). They had a mean age of 55.8 (*SD* = 5.4), with children with a mean age of 25.3 (*SD* = 5.6), a mean of 1.7 children (*SD* = 0.7), with a mean degree of disability of 47.8 (*SD* = 15.3). Most of the children (83.34%) had a diagnosis of mild intellectual disability, and the rest (16.66%) had medical conditions (e.g., Down’s Syndrome or Fetal Alcohol Syndrome) associated with intellectual disability. The sociodemographic characteristics of the study can be obtained from the first author of the manuscript.

### 2.3. Measures

The brief intervention protocol (9 h) followed a similar structure to the programme developed by Whittingham et al. (2014). It focused on (a) values clarification, (b) defusion strategies, (c) training in flexible attention to the present moment (mindfulness), and (d) commitment to action and psychological acceptance. The therapeutic methods were adapted from theoretical and clinical skills manuals published by [30,31,32]. Table 1 lists the contents of the intervention (a more detailed version of the protocol is available on request from the first author).

The evaluation instruments were administered as follows:

*Parental Acceptance Questionnaire (6-PAQ)* [33]. To evaluate parental PF, the Spanish version of 6-PAQ was used [34]. It is a 16-item questionnaire on a Likert-type scale with four answer options in a range from 1 (Strongly Disagree) to 4 (Strongly Agree) that assesses six processes related to PF (being present, values, committed action, self as context, cognitive defusion, and acceptance) and three flexible response styles (opened, centred and committed). The scores vary from 16 to 64; the higher the score, the higher the psychological inflexibility (PI) levels. In studies on psychometric properties of the scale, Cronbach’s alpha scores of 0.81 and McDonald’s omega of 0.86 were reported.

*Perceived Stress Scale (PSS-14)* [35]. The Spanish adaptation of PSS-14 [36] was used. It is a widely used measure to assess the degree of perceived control over life circumstances. In the study on psychometric properties, Cronbach’s alpha values of 0.72, 0.82, and 0.86 were obtained. It is a one-dimensional scale with 14 items that are answered on a Likert-type scale ranging from 0 (Never) to 5 (Very often). The direct scores range from 0 to 56; higher scores indicate higher perceived stress.

*Psychological Health Questionnaire (GHQ-12)* [37]. It is a widely used measure for the assessment of psychological health. It contains 12 items, and its consistency is measured with Cronbach’s alpha of 0.85. Higher scores indicate lower levels of psychological well-being. The Spanish-validated version was used, showing a Cronbach alpha of 0.76 [38].

*White Bear Suppression Inventory (WBSI)* [39]. The Spanish validation of WBSI [40] was used to evaluate the tendency to suppress unwanted thoughts. It is a Likert scale of 15 items with five response options ranging from 1 (Completely Disagree) to 5 (Completely Agree). Scores range from 15 to 75. Higher scores indicate a stronger thought suppression tendency. Regarding the internal consistency of the scale, Cronbach’s alpha values of 0.89 were reported in the general scale, and values of 0.87 and 0.80 for its subscales.

*Behavioural self-monitoring.* To capture behaviour change, family members recorded a daily estimation of the frequency of two categories of behaviours: punitive-hostile behaviours from family members to children with disabilities (e.g., shouting, punishments, insults, or aggressions) and supportive-companion behaviours (e.g., helping, shared leisure and recognition or compliments). A Likert-type scale was used to estimate frequency, with values from 0 to 4 (0 = never; 1 = almost never; 2 = sometimes; 3 = often; and 4 = always). Participants completed daily self-monitoring, starting one week before the intervention (forming the baseline scenario: BL onwards) until one week after the intervention, obtaining a total of 4 measures.

**Table 1 ijerph-19-13943-t001:** Description of the clinical trial intervention protocol.

Therapeutic Processes							Sessions		
Therapeutic Methods	Acceptance	Defusion	Values	Committed Action	Contact with Present	Self as Context	1	2	3
Promoting creative hopelessness: an experiential exercise to assess the effectiveness of stress coping repertoires.	X						X		
“Man in the hole” metaphor [26]									
“The compass metaphor” [41]			X	X				X	
Experiential exercise: What kind of mother or father do I want to be?									
Experiential exercise: What can I do this week to move towards the kind of parent I want to be?									
Observer exercise [26]						X	X	X	X
Homework: implementing committed actions and self-report.				X			X	X	X
Exercise: Clarifying values about family/parenting; assessing the importance and value orientation on a scale of 0 to 10.			X					X	
Experiential Exercise: Emotion Mindfulness through physicalization [42]		X					X		
“Joe the Bum” metaphor [26]									
“Passengers on the Bus” metaphor [26]		X						X	X
Experiential exercise: Mindful breathing [42]					X			X	X
Homework: Practicing mindful breathing exercises (audio recording)							X	X	X
Experiential exercise: Our values as parents. Sharing the values on which I articulate my actions.			X						
Identifying valuable actions in the presence of internal and external barriers. Experiential exercise: “When… Then”.									X
Experiential exercise: “The choice point” [42]									X

### 2.4. Procedure

The application of the brief intervention protocol was implemented over three sessions of 3 h each week in a room of the collaborating NGO by a psychologist trained in third-generation therapies and experienced in working with families (main author). Self-reports were administered before the intervention (pre-treatment), one week after its completion (post-treatment), and two months later (follow-up). Participants completed daily self-monitoring from one week before the intervention (forming the LB) until one week after the intervention, obtaining 4 measures. In addition, the participants filled in their workbooks in each session, carried out the programmed activities, voluntarily participated, and signed the informed consent form.

### 2.5. Statistical Analysis

A descriptive analysis of the variables evaluated at each of the three defining moments was carried out: pre-treatment, post-treatment, and follow-up (2 months). In addition, two complementary strategies were used to evaluate the intervention protocol’s effectiveness. First, the participant scores on each of the instruments before the intervention programme were compared with the scores obtained after the programme (i.e., pre-post) and two months later (i.e., pre-follow-up). For this purpose, a Student’s *t*-test was applied to each of the comparisons performed, and the effect size was estimated by calculating Cohen’s d [43]. Second, the clinical significance of the changes obtained as a result of the application of the protocol was also contrasted using the method proposed by Jacobson & Truax [44] for each of the variables considered. The authors propose identifying a cut-off score to determine whether it is clinically significant. If the change was more than one standard deviation away from the average distribution of scores of family members of people with disabilities who participated in the study, it is assumed to be clinically significant [45].

In addition, to determine whether this observed change is reliable, the authors proposed a reliable change index (RCI) based on the estimation of the standard error of the difference between the pre-and post-intervention scores. It considers a change in a particular participant to be reliable when it exceeds the width of the distribution of change scores that would be expected if no change had occurred as a result of the intervention. Since the RCI is stated in the number of standard deviations, scores higher than 1.96 will be interpreted as reliable changes, assuming a probability of error of 0.05. From these two values, the cut-off points and the RCI, the authors propose to estimate whether the patient will have recovered, improved, shown no change (i.e., no change), or worsened as a result of the intervention, according to the following criteria. The patient will have recovered if the RCI value is more significant than 1.96 and the observed change exceeds the cut-off point. Likewise, the patient will have improved if the RCI value is greater than 1.96, but the observed change does not exceed the cut-off point. Lastly, the patient will show no change (i.e., no change) if the RCI value is less than or equal to 1.96 and will have worsened; or if the RCI value is greater than 1.96 but in the opposite direction to functionality.

## 3. Results

The programme showed strong adherence among the participants who attended the training sessions, completed the assessments, and filled in the self-monitoring daily. Table 2 shows the means and standard deviations obtained in each of the scales applied and in the subscales that comprise them, recorded at the above-mentioned three points in time. Likewise, the table shows the results of the comparisons carried out using the Student’s *t*-test and the effect size.

A decreasing trend in psychological inflexibility, perceived stress, psychological distress, and suppression of private events were observed in participants’ average total scale and subscale scores before and after the protocol implementation. All mean scores on the main scales and subscales showed statistically significant changes in the expected direction, except for the pre-post evaluation of the 6-PAQ subscale (actions) and the pre-post evaluation of the PSS-14 and GHQ-12 scales. In all cases, and regardless of statistical significance, the scales and subscales offered medium to large effect sizes. The same results were observed when comparing the mean difference before the protocol implementation with the two-month follow-up phase.

An interesting observation is that, as time goes by, participants’ performance results consolidate or improve, except for the 6-PAQ subscale (acceptance). The statistical differences observed between the phases prior to and following the protocol application were maintained, and statistical significance was reached in those tests that did not reach it before (“actions” subscale of the 6-PAQ, PSS-14 and GHQ-12). Again, medium and high effect sizes were observed when comparing these two evaluation time points. Furthermore, the results show that the observed changes persisted at least two months after the protocol implementation, and the intervention effect increased over time.

Regarding the analysis of the clinical significance measures, Table 3 presents the change data through the participants’ scores before and immediately after the application of the intervention protocol (pre-post) and before and two months after it (pre-follow-up) for each of the applied questionnaires. The percentages represent the total number of participants for each condition established by Jacobson and Truax [44].

The intervention programme was followed by a behavioural change in the expected direction (improved or recovered) on the following variables: on the 6-PAQ (psychological flexibility) in 63.8% of the participants at the post and 66.6% at follow-up; on the PSS (perceived stress), in 36% of participants at the post and 61.1% at follow-up; on the GHQ (psychological health), in 25% at the post and in 55.5% at follow-up; on the WBSI (suppression of unwanted private events), in 52.7% of participants at the post and 58.3% at follow-up. The observed change involved an increase in psychological flexibility, a decrease in perceived stress, an improvement in psychological health, and a tendency to decrease the suppression of unpleasant private events after the programme. The effect size was medium-large for all main scales and subscales. Changes were sustained, and participants continued to improve during the follow-up phase, especially in the stress and health variables.

Finally, the change in the parent-child interaction responses (supportive/companioning and punitive/hostile behaviours) was analysed across the 4-time points (BL, W1, W2, and W3) using repeated measures ANOVA. Concerning supportive/companion interactions, a notable increase in responses was observed from BL to the end of W3, while punitive/aversive interactions showed a decrease from BL to the end of W3. Table 4 shows the means and standard deviations obtained at each time stage. Figure 1 shows the upward change in the response tendency, represented on the *Y*-axis by the mean frequency of participants’ responses (for *n* = 36).

A statistically significant change followed the intervention programme in interaction repertoires for both supportive/accompanying behaviours (F (3.105) = 26.692; *p* < 0.001) and punitive/hostile behaviours (F (3.105) = 27.564; *p* < 0.001) in the direction expected as seen in Table 5.

As shown in Table 6, the size of the support/accompaniment behaviour changes is medium to large, being more significant when analysing the difference between the BL and the last week of intervention, where an upward trend is observed.

Regarding punitive/hostile behaviours, as shown in Table 7, a decrease in responses is observed from BL to the end of W3. The magnitude of the changes is medium to large, except for the behavioural change between W2 and W3, which is smaller. Similar to the support/companionship interactions, the most significant changes happened between the BL and the last week of intervention, where a decreasing trend was observed.

## 4. Discussion

The ACT-based intervention for relatives of children with disabilities was followed by a significant reduction in psychological inflexibility, thought suppression, and stress and an increase in psychological well-being. Changes in the variables of psychological flexibility and private events suppression are statistically significant at both post-intervention and follow-up. At follow-up, statistical significance is reached for all variables studied. Furthermore, supportive-companion interaction responses increased, and hostile-punitive interaction responses towards the children statistically significantly decreased. The effect size was large-medium for all scales and subscales. Moreover, the intervention administration in the usual family care context supports the ecological validity of the intervention. The results are similar to those found in recent ACT studies among caregivers of people with intellectual disabilities [46]. These results are consistent with those obtained in the pilot study conducted by the research group, where the psychological flexibility pilot programme reduced psychological inflexibility and thought suppression. However, in contrast to this study, there were no significant changes in stress levels and psychological well-being.

The analysis of process measures through RCI allowed us to conduct a more idiographic and descriptive analysis of the behavioural changes. A significant and more immediate effect of the programme was observed on the variables related to psychological flexibility and suppression of unpleasant private events than on the variables related to stress and psychological health, which reached moderate effectiveness during the follow-up phase, in congruence with the ACT hypotheses [1]. Persons with disabilities’ relatives live with stressors that cannot be changed or solved, either because of the chronic nature of the difficulties associated with the disability or because behavioural change is slower than in the group of people without disabilities, to the extent that other skills and competences are required to cope with such situations. Thus, ACT exercises are designed to train people to more functionally and flexibly deal with their daily stressors [47] and appear more effective in the medium to long term. This is one of the fundamental differences from CBT treatments for parental stress, which showed effects post-treatment, but the effect faded over time [48].

The results obtained in the study provide evidence for the administration of the programme to parents in a group format because family members can act as role models among themselves, creating a support network among family members and especially among mothers [49].

Most behavioural interventions aimed at parents seek to incorporate more functional repertoires to manage the behaviour of children with disabilities. However, ACT interventions not only focus on the parents’ overt behaviour but also on improving their well-being and health, providing them with strategies to interact more flexibly (through their different processes) with their private events, orienting them towards valuable parenthood. Therefore, our programme targets the stress of parenting a child with a disability, so improving the relationship with unpleasant private events can also affect the diagnosed child’s behaviour [50].

A cyclical relationship exists between parental stress and the child’s behavioural problems [51], so breaking this cycle becomes a clinical priority in treating children with disabilities. The incorporation of stress reduction programmes prior to or concurrent with behaviour modification training for parents has summative effects on outcomes [52]. For example, the meta-analysis by [53] suggests that stress coping strategy training and the behavioural management programme for parents produces enhancement effects on the behavioural intervention. Therefore, parents’ involvement (and their associated suffering) in the behavioural intervention of a child with a disability seems crucial. Along these lines, the observed changes in interactions with children show an increasing pattern of supportive/accompanying behaviours and a decreasing pattern of punitive/hostile behaviours. These results are similar to those in the scientific literature [54]. Following this line of research, the circle of hostility in interactions would be weakened, and parents would respond with less punitive behaviour, resulting in behavioural and emotional improvement for the diagnosed child. Coaching the relevant people to clarify their values, from which they wish to orient parenting, can be tremendously productive. As these people undertake committed actions (verbally regulated), they will likely be reinforced by the direct consequences provided by the context, such as the children’s improvement, an increase in the family system’s own shared joy, and a strengthening of family relationships.

On the other hand, this clinical trial still has significant limitations. First of all, those derived from the sampling. The number of participants was increased to generalise the results obtained and avoid the likelihood of Type II errors. However, adding an even larger number of participants would make the results more generalisable. In addition, the participants were primarily mature women with older children, for whom the training seems to be particularly effective. However, future research should study whether this same programme works in a sample of mostly fathers, as there are programmes in the literature that do not show the same effects on them [55]. Secondly, regarding the experimental design, the absence of a control group in the study could limit the validity of the results. In our research, we decided not to carry it out for ethical and pragmatic considerations and to fit the NGO’s care needs. However, aware of this limitation, our research group is finalising a randomised clinical trial with a control group to overcome this limitation. Thirdly, the severity of the disability, the type of impairment, or the behavioural manifestations may affect the results and distort the intervention program’s differential efficacy or the size of its effects [56]. For this reason, in future studies, it would be interesting to consider the diagnosis or to carry out an individualised subject-by-subject analysis to determine the effect of the programme, depending on the type of diagnosis. Fourthly, the same researcher who designed the research was the one who applied the intervention protocol and did the data analysis. It would be interesting for future research if the therapist was a different person, trying to eliminate observer bias or the placebo effect as much as possible.

Fifth, as in other trials assessing change in interactions following parent training programmes, our trial used a self-report (subjective) measure and no prior training in behavioural observation, which may have affected the reliability of the data. Besides the use of these instruments, the addition of objective measures or standardised questionnaires (e.g., the CBCL: Child Behaviour Checklist) and/or prior training in the correct recording of interactions could be a factor in the improvement of the study. Sixthly, it would be interesting to add an assessment of interactions during the follow-up phase (e.g., at 6 or 12 months) to see whether the repertoires registered post-intervention are maintained over time, as one of the criticisms of traditional behaviour modification programmes for families is their diminishing effect in the long term [57].

Finally, this study provides new scientific evidence about the role of psychological flexibility as a relevant variable for improving the health of parents facing chronic stress situations, such as raising a child with a disability and building more conscious and effective parenting.

## 5. Conclusions

The program results suggest a decrease in psychological inflexibility, reduced perceived stress, improved psychological well-being, and reduced tendency to suppress thoughts post-intervention, especially at the follow-up. However, the most substantial effects of the group intervention protocol are still shown in psychological flexibility and suppression of private events. The above results lead us to think the intervention in psychological flexibility trains skills to respond more flexibly to aversive private events derived from raising a child with a disability. It may promote the decrease of suppressive behaviour and orients the interaction towards the values as a parent, focusing on medium and/or long-term reinforcement. Regarding the changes observed in the interaction with children diagnosed with disabilities, in the present study, there is a significant increase in supportive/companioning interactions and a significant decrease in punitive/hostile interactions after the second week of training. Therefore, it appears that training in psychological flexibility implies an orientation of the parent towards their parental values, delimits what type of interaction they wish to cultivate with their children, and teaches, through various techniques (e.g., defusion and/or acceptance), to overcome the barriers that arise when their behaviour is oriented towards their children with disabilities. The above results in the implementation of less hostile and more therapeutic behaviours, reducing the feedback loops of aggression. This effect has been particularly striking because the intervention protocol did not include specific skills training for managing disruptive behaviour. Therefore, unlike traditional behaviour modification programs that focus on specific behavioural management skills, parents’ psychological flexibility training could produce summative and extensive effects in such programs.

## Figures and Tables

**Figure 1 ijerph-19-13943-f001:**
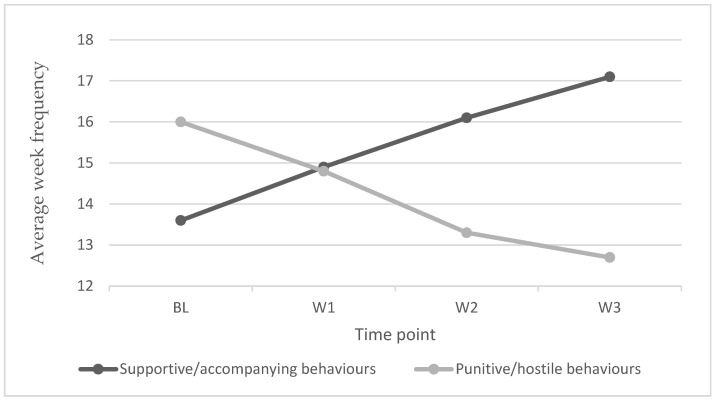
Evolution of family interactions. (Note. Supportive/accompanying and punitive/hostile interactions over the three weeks the intervention was implemented).

**Table 2 ijerph-19-13943-t002:** Questionnaire scores and effect comparisons in the clinical trial.

	Pre	Post	Follow-Up	Pre-Post			Pre-Follow-Up		
Variables	M (*SD*)	M (*SD*)	M (*SD*)	*t*	*p*	*d*	*t*	*p*	*d*
6-PAQ-Total	42.4 (6.7)	36.2 (4.4)	34.8 (4.6)	7.12	<0.001	1.18	7.16	<0.001	1.19
6-PAQ-Acceptance	7.3 (1.5)	6.4 (1.2)	6.4 (1.1)	3.81	<0.001	0.63	3.49	0.001	0.58
6-PAQ-Defusion	8 (1.6)	6.5 (1.1)	6.5 (1.2)	7.18	<0.001	1.19	5.56	0.001	0.92
6-PAQ-Being present	5.5 (1)	4.4 (0.8)	4.1 (0.6)	6.53	<0.001	1.08	7.69	<0.001	1.28
6-PAQ-Values	5.3 (1.1)	4.5 (0.84)	4.4 (0.7)	4.93	<0.001	0.82	5.01	<0.001	0.83
6-PAQ-Committed action	7.9 (1.6)	7 (1.4)	6.8 (1.2)	3.06	0.004	0.51	3.78	<0.001	0.63
6-PAQ-Self-as-context	8.2 (1.3)	7 (1.2)	6.2 (1.2)	5.30	<0.001	0.88	6.41	<0.001	1.06
PSS-Total	31.8 (6.1)	30.1 (4.3)	28 (4.3)	2.85	0.007	0.47	3.99	<0.001	0.66
GHQ-Total	26 (6.7)	25.4 (6.1)	21.8 (6.6)	2.54	0.015	0.42	6.5	<0.001	1.08
WBSI-Total	45.6 (11)	40.3 (8)	36.8 (6.4)	6.24	<0.001	1.04	7.17	<0.001	1.19

Note. Mean scores and standard deviations, preintervention, postintervention and follow-up (*n* = 36), Student’s t mean comparison statistic (*t*) and effect size (*d*), M = mean; *SD* = standard deviation; 6-PAQ = psychological inflexibility measured via the Parental Acceptance Questionnaire; PSS = perceived stress measured through the Perceived Stress Scale; GHQ = psychological distress measured through the General Health Questionnaire; WBSI = thought suppression measured through the White Beard Suppression Inventory.

**Table 3 ijerph-19-13943-t003:** Clinical significance of changes observed as a result of the implementation of the intervention programme for *n* = 36 (%).

Pre-Post					Pre-Follow Up			
	Wor	NCS	Imp	Rec	Wor	NCS	Imp	Rec
PAQ	0 (0%)	13 (36.1%)	13 (36.1%)	10 (27.7%)	0 (0%)	12 (33.3%)	7 (19.4%)	17 (47.2%)
PSS	0 (0%)	23 (63%)	10 (27.7%)	3 (8.3%)	0 (0%)	14 (38.8%)	15 (41.6%)	7 (19.4%)
GHQ	0 (0%)	27 (75%)	9 (25%)	0 (0%)	0 (0%)	16 (44.4%)	19 (52.7%)	1 (2.7%)
WBSI	0 (0%)	17 (47.2%)	18 (50%)	1 (2.7%)	0 (0%)	15 (41.6%)	15 (41.6%)	6 (16.6%)

Note: Group results obtained as a result of the intervention protocol application: Wor: Worsened; NCS: No clinically significant change; Imp: Improved; Rec: Recovered.

**Table 4 ijerph-19-13943-t004:** Descriptive statistics for supportive/accompanying and punitive/hostile behaviours.

Supportive/Accompanying Behaviours			Punitive/Hostile Behaviours		
Time	M	*SD*	Time	M	*SD*
BL	13.6	1.9	BL	16	2.4
W1	14.9	1.5	W1	14.8	1.6
W2	16	2	W2	13.3	1.5
W3	17.1	2	W3	12.7	1.6

**Table 5 ijerph-19-13943-t005:** Programme statistical significance on parenting behaviours.

Cases	*gl*	Root Mean Square	F	*p*
Supportive/accompanying behaviours	3	79.611	26.629	<0.001
Residuals	105	2.983		
Punitive/hostile behaviours	3	81.322	27.564	<0.001
Residuals	105	2.950		

Note. *gl* = degrees of freedom; F = Snedecor’s F; *p* = statistical significance.

**Table 6 ijerph-19-13943-t006:** Size effect of the programme.

Evaluation Timing		Mean Difference	*t*	*d* Cohen	*p* Holm
BL	W1	−1.333	−3.276	−0.546	0.004 **
	W2	−2.417	−5.937	−0.989	<0.001 ***
	W3	−3.472	−8.530	−1.422	<0.001 ***
W1	W2	−1.083	−2.661	−0.444	0.018 *
	W3	−2.139	−5.254	−0.876	<0.001 ***
W2	W3	−1.056	−2.593	−0.432	0.018 *

Note. *t*: Student’s *t*, *d*: Cohen’s *d*, *p*: *p* Holm, * *p* < 0.05, ** *p* < 0.01, *** *p* < 0.001.

**Table 7 ijerph-19-13943-t007:** Programme effect size on negative interactions.

Evaluation Timing		Mean Difference	*t*	*d* Cohen	*p* Holm
BL	W1	1.250	3.088	0.515	0.005 **
	W2	2.694	6.655	1.109	<0.001 ***
	W3	3.361	8.302	1.384	<0.001 ***
W1	W2	1.444	3.568	0.595	0.002 **
	W3	2.111	5.214	0.869	<0.001 ***
W2	W3	0.667	1.647	0.274	0.103

Note. *t*: Student’s *t*, *d*: Cohen’s *d*, *p*: *p* Holm, ** *p* < 0.01, *** *p* < 0.001.

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
