# Peer review of "Acceptance and Commitment Training Focused on Psychological Flexibility for Family Members of Children with Intellectual Disabilities"

_ijerph, 2022, doi:10.3390/ijerph192113943_

Round 1

Reviewer 1 Report

Dear authors, congratulations on your article. It's an article of great interest. However, I would like to point out the following:

The research paper, “ Acceptance and Commitment Training Focused on Psychologi- 2 cal Flexibility for Family Members of Children with Intellec- 3 tual Disabilities “ is a very important research that examinesthe effect of a psychological flexibility intervention program on family members of children with intellectual disabilities with the importance of psychological flexibility in children with chronic diseases and as a process that mediates the impact of stress and family well-being that affect the quality of life of a family with a child with intellectual disabilities.

The whole research article is well-organized, informative, and easily readable for everybody to understand the theoritical background and the methology.

The Abstract is concise, and it is consistent with the content of the main text.

The Introduction section has the relevant information but i do have one comment here. It would have been better if the authors had devoted 1-2 paragraphs to referbut the impact it has on their families, both in terms of the cohesion and relationship that develops between its members and its emotional intelligence to cover the theoretical part from all angles. That would strengthen the article. I think LITERATURE REVIEW needs more documentation and bibliography review. Possible supporting literature would be the following:

1. Chaidi , I. ., Drigas, A., & Karagiannidis, C. (2021). Autistic people’s family and emotionalintelligence. Technium Social Sciences Journal, 26(1), 194–214

2.  N. Luitwieler, J. Luijkx, M. Salavati, C.P. Van der Schans, A.J. Van der Putten, A. Waninge, Variables related to the quality of life of families that have a child with severe to profound intellectual disabilities: A systematic review, Heliyon, Volume 7, Issue 7, 2021, e07372, ISSN 2405-8440, https://doi.org/10.1016/j.heliyon.2021.e07372.

(https://www.sciencedirect.com/science/article/pii/S2405844021014754)

3.  Helen M. Bourke-Taylor, Loredana Tirlea, Kahli S. Joyce, Further psychometric evaluation of the My Family’s Accessibility and Community Engagement (My FACE) tool: Mothers’ ratings of perceptions of community accessibility and engagement for their child with disabilities, Research in Developmental Disabilities,Volume 114, 2021, 103955, ISSN 0891-4222, https://doi.org/10.1016/j.ridd.2021.103955.

(https://www.sciencedirect.com/science/article/pii/S0891422221001049)

4. A.Bhopti, T. Brown, P. Lentin, Family quality of life: a key outcome in early childhood intervention services–A scoping reviewJ. Early Interv.,38 (4) (2016), pp. 191-211

The Method section presents all the details about the design, the search strategy, the criteria etc.

The Method section presents all the details about the design, the search strategy, the criteria etc.

The Results section indicate a good effort and provide the necessary statistical power to the study. Table 2, 3, 4, 5, 6, 7 and Figure 1 are appropriate and very enlightening especially the second one with the variables.

The Discussion section gathers all the information together into a single whole. Authors address important aspects of the research and provide specific explanations about the main findings, theoretical and methodological reflections and recommendations for future research related to effect of a psychological flexibility intervention program on family members of children with intellectual disabilities with the importance of psychological flexibility in children with chronic diseases and as a process that mediates the impact of stress and family well-being.

We suggest to enhance the bibliography with similar articles as the one that follows in order to take into account the role of emotional intelligence.

Concluding we underline that this is a promising article but it needs more documentation and literature enhancement, and we suggest that as far as the authors will make the noticed improvements, the article could be published in MDPI journal

Author Response

Response to Reviewer 1 Comments

Point 1: It would have been better if the authors had devoted 1-2 paragraphs to refer but the impact it has on their families, both in terms of the cohesion and relationship that develops between its members and its emotional intelligence to cover the theoretical part from all angles. That would strengthen the article. I think literature review needs more documentation and bibliography review.

Response: We have reviewed the bibliography you sent us and added some of the authors you recommend, emphasizing the importance of family intervention in the context of disability.

Point 2: Concluding we underline that this is a promising article but it needs more documentation and literature enhancement

Response: We have reviewed the literature and added five references to update and strengthen the justification of the intervention program.

Reviewer 2 Report

Dear authors,

I consider that it is a correctly written article and therefore suitable for publication in the IJERPH journal.

Only two observations:

-          I think would be appropriate to use the acronym PSS-14 for the scale mentioned in the manuscript (you used the 14 items version);

-          For me, it is not clear how did you group the subjects into groups. Were all the subjects who withdrew initially assigned to group 4?

Author Response

Response to Reviewer 2 Comments

Point 1:  I think would be appropriate to use the acronym PSS-14 for the scale mentioned in the manuscript (you used the 14 items version)

Response: We have replaced the acronym PSS by PSS-14.

Point 2: For me, it is not clear how did you group the subjects into groups. Were all the subjects who withdrew initially assigned to group 4?

Response: Yes. We made groups of 10 family members. Unfortunately, group 4 lost four family members (n=6) before starting the intervention. We understand that this may be because the start of the treatment was delayed in time, and the subjects lost motivation or found it impossible to attend.